# Co-Expression of Coxsackievirus/Adenovirus Receptors and Desmoglein 2 in Lung Adenocarcinoma: A Comprehensive Analysis of Bioinformatics and Tissue Microarrays

**DOI:** 10.3390/jcm9113693

**Published:** 2020-11-18

**Authors:** Ching-Fu Weng, Chi-Jung Huang, Mei-Hsuan Wu, Henry Hsin-Chung Lee, Thai-Yen Ling

**Affiliations:** 1Division of Pulmonary Medicine, Department of Internal Medicine, Hsinchu Cathay General Hospital, Hsinchu 300, Taiwan; philweng11@yahoo.com.tw; 2Department and Graduate Institute of Pharmacology, National Taiwan University, Taipei 100, Taiwan; 3Medical Research Center, Cathay General Hospital, Taipei 106, Taiwan; aaronhuang@cgh.org.tw; 4Department of Biochemistry, National Defense Medical Center, Taipei 114, Taiwan; 5School of Medicine, Fu Jen Catholic University, New Taipei 242, Taiwan; 6Teaching and Research Center, Hsinchu Cathay General Hospital, Hsinchu 300, Taiwan; s101082015@gapp.nthu.edu.tw; 7Department of Surgery, Hsinchu Cathay General Hospital, Hsinchu 300, Taiwan; 8Graduate Institute of Translational and Interdisciplinary Medicine, College of Health Sciences and Technology, National Central University, Taoyuan 320, Taiwan

**Keywords:** coxsackievirus and adenovirus receptor, desmoglein-2, lung adenocarcinoma, immunohistochemistry, prognosis prediction

## Abstract

Introduction: Coxsackievirus/adenovirus receptors (CARs) and desmoglein-2 (DSG2) are similar molecules to adenovirus-based vectors in the cell membrane. They have been found to be associated with lung epithelial cell tumorigenesis and can be useful markers in predicting survival outcome in lung adenocarcinoma (LUAD). Methods: A gene ontology enrichment analysis disclosed that DSG2 was highly correlated with CAR. Survival analysis was then performed on 262 samples from the Cancer Genome Atlas, forming “Stage 1A” or “Stage 1B”. We therefore analyzed a tissue microarray (TMA) comprised of 108 lung samples and an immunohistochemical assay. Computer counting software was used to calculate the H-score of the immune intensity. Cox regression and Kaplan–Meier analyses were used to determine the prognostic value. Results: *CAR* and *DSG2* genes are highly co-expressed in early stage LUAD and associated with significantly poorer survival (*p* = 0.0046). TMA also showed that CAR/DSG2 expressions were altered in lung cancer tissue. CAR in the TMA was correlated with proliferation, apoptosis, and epithelial–mesenchymal transition (EMT), while DSG2 was associated with proliferation only. The Kaplan–Meier survival analysis revealed that CAR, DSG2, or a co-expression of CAR/DSG2 was associated with poorer overall survival. Conclusions: The co-expression of CAR/DSG2 predicted a worse overall survival in LUAD. CAR combined with DSG2 expression can predict prognosis.

## 1. Introduction

For decades, lung cancer has been recognized as the leading cause of cancer-related deaths worldwide. In general, around 80% of all lung cancers are non-small-cell lung cancers (NSCLCs), which comprise lung adenocarcinoma (LUAD), squamous cell carcinoma (LSCC), and large-cell carcinoma. Small-cell lung cancers account for only 10% to 15% of all lung cancers but are typically more aggressive [1,2]. The 5-year survival for all types of lung cancer is around 19%, yet the majority of cases are diagnosed at an advanced stage, which has a poor 5-year survival rate of <10% [3]. The development of druggable genome targets and immunotherapy have notably improved prognostic outcomes in recent years, but the survival rate remains low. Early diagnosis with prompt intervention is crucial to decrease lung-cancer-associated mortality, which would make a tool that could detect, early on, prognostic predictions very valuable.

The coxsackievirus and adenovirus receptor (CAR) is a tight-junction protein localized to the cell membrane as part of the immunoglobulin (Ig) superfamily. It is associated with cell–cell adhesions as part of the apical junctional complex [4]. Previous studies indicated that CAR plays a key role in mediating gene delivery via therapeutic adenoviral vectors into tumor cells. Increased CAR expression is predictive of more efficient gene transfer into lung cancer cells, both in vitro and in vivo [5,6]. Elevated CAR expression was detected in malignant tumors from breast cancer, lung cancer, and several female reproductive system cancers [7,8,9,10], while a reduced expression was observed in gastric cancer, colorectal cancer, and prostatic and kidney neoplasms [11,12,13,14,15]. Overexpression of CAR is considered to promote carcinogenesis when it occurs in breast cancer precursor cell lines [16]. Acquisition of tumorigenic potential has been demonstrated in a mouse lung progenitor cell model after sorting for CAR as a positive selection marker [17]. CAR is a potential marker of cancer stem cells (CSCs) and caused drug resistance in NSCLC [18]. In short, high CAR expression might be a treatment failure predictor in NSCLC, indicating a poor survival outcome in lung cancer [7,9,18].

There are four desmoglein (DSG) proteins—DSG1, DSG2, DSG3, and DSG4—that are members of the cadherin family, which are transmembrane proteins that bind with other cadherins. DSG2 participates in cell growth, proliferation, apoptosis, and tumorigenic migration, as well as in invasion [19,20]. A previous study used hDSG2-transgenic mice infected with a green fluorescent protein (GFP)-expressing HAdV-B3 vector (Ad3-GFP). The efficient transduction of bronchial and alveolar epithelial cells was detected by an intranasal spray. DSG2-interacting Ad3-derived vectors in gene therapy were used in the animal model [21]. DSG2 is also expressed in various malignant tumors and is correlated with the risk of metastasis and inferior outcomes [22,23,24]. The prognostic value of DSG2 has been previously addressed in NSCLC [25,26]. One study demonstrated that DSG2 expression was associated with tumor size, lymph node metastasis, and TNM (TNM staging system), as well as being a predictor of poor prognosis in LUAD [27].

CAR and DSG2 are similar molecules and can be recognized by adenovirus-based vectors in the cell membrane [28]. Further, they can be useful markers in predicting survival outcome in lung cancer, as mentioned earlier. To the best of our knowledge, there has not been a comprehensive study of the potential roles of DSG2 and CAR in oncolytic therapy for lung cancer or of the correlation between their high expression and survival. Therefore, the current study investigated the use of CAR in combination with DSG2 as a predictor of LUAD outcomes.

## 2. Materials and Methods

### 2.1. Data from Online Databases

To investigate gene expression in lung cancer, four public domain datasets were collected from the gene expression omnibus (GEO) from the National Center for Biotechnology Information. The accession numbers were GSE102511, GSE32863, GSE52248, and GSE68571 (Appendix A). Datasets GSE102511 and GSE52248 were gene expression data for normal lung parenchyma (NL) tissue that progressed to cancer, while GSE32683 and GSE68571 were gene expression data that compared NL tissue with cancerous tissue. CAR was upregulated in the normal samples that progressed to cancer in GSE102511 and GSE52248. Only Stage I samples were selected in the GSE32863 and GSE68571 datasets. The information of files used to do analysis on these four datasets is listed in Appendix A
Appendix A. Figure 1 demonstrated the analysis flow to identify two groups of genes on *CAR* gene sequential changes from NL: atypical adenomatous hyperplasia (AAH) to invasive LUAD. One group was the same expressed trend genes as *CAR* gene. Another group was the contrary expressed trend as *CAR* gene. The genes of at least three datasets that had the same or contrary expressed trends as *CAR* gene were selected in each group. Then, these two groups of genes were screened to only include cancer-related genes. The ClusterProfiler R package was used to perform gene function analysis for the two groups of cancer-related genes [29]. Gene interaction network analysis was performed using the Search Tool for the Retrieval of Interacting Genes/Proteins (STRING, www.string-db.org).

To assess a survival analysis associated with the two groups of gene expression, 594 LUAD RNA-seq data were obtained from the the Cancer Genome Atlas (TCGA, http://cancergenome.nih.gov). “FPKM-UQ” was set as the measurement unit of the gene expression level. According to the clinical information from metadata of 594 samples, 262 samples that were “Stage 1A” or “Stage 1B” were included for further analysis. The expression of CAR was divided by the average expression level of the housekeeping genes from each sample. Two group of genes expressed either the same or contrary trends as *CAR* gene, which were also used to identify additional potential biomarkers. *DSG2* gene was found to be highly correlated with *CAR* gene. In the 262 samples, the expression levels of *CAR* and *DSG2* gene were defined as high if their expression was >90% of the samples. Survival analysis of the 262 samples was performed by the R package “survminer”.

### 2.2. Patients and Tissues

CAR and DSG2 protein expression were evaluated using tissue microarrays (TMAs). The TMA (catalogue number L-LT-F-01-A, L-LT-F-02-A) was comprised of 108 lung cancer formalin-fixed samples, paraffin-embedded samples, and two normal adjacent non-tumor tissues, which were purchased from Tech Smart Hong Kong, Ltd., Hong Kong. All human samples met the requirements of the Human Material Transfer Agreement (MTA). TMA patients consisted of 63 males and 45 females, with a median age of 67.1 years (range, 36–87 years). All patients had follow-up records for >10 years. The survival time was calculated from the date the specimen was obtained to the follow-up deadline or mortality. Post-mortem specimens and surgically obtained tissues were collected under ethical conditions, and both the donor and their relatives (where appropriate) provided informed consent. All tissue samples underwent standard medical care to protect the donor’s privacy. Informed consent was recorded and retained by the tissue banks of the certified hospitals that obtained the tissue. All enrolled patients were de-linked anonymously for protection. This study protocol was approved by the institutional review board of the institution.

### 2.3. Statistical Analysis

Statistical analysis was performed using SPSS version 20.0 (IBM Corp, Armonk, NY, USA). Correlations between two categorical variables were examined using Pearson’s chi-square test. Progression-free survival (PFS) and overall survival (OS) were calculated at 12 and 36 months after the initial diagnosis, respectively. The Kaplan–Meier method was used to analyze the distributions of PFS and OS, and log-rank tests were performed to compare the differences between the two categories. Univariate analysis and multivariate survival analysis were conducted using the Cox proportional hazards regression model to obtain hazard ratios (HRs) with 95% confidence intervals (CIs), as well as to identify independent prognostic factors for PFS and OS. Statistical significance was set at α = 0.05.

### 2.4. Immunohistochemical Staining (IHC)

IHC assays were completed according to the standard method. Array slides were shipped at room temperature. All tissue array slides were covered with an extra layer of paraffin and were stored at 4 °C. The slides were heated at 60 °C for at least 30 min before de-paraffinization via xylene. The tissue blocks were cut into sections for IHC stain. The IHC stain was performed using a Ventana BenchMark XT automated stainer (Ventana, Tucson, AZ, USA). Briefly, 4 μm-thick sections were cut consecutively from formalin-fixed tissue and paraffin-embedded tissue. Sections would be mounted on salinized slides and allowed to dry overnight at 37 °C. After deparaffinization and rehydration, slides would be incubated with 3% hydrogen peroxide solution for 5 min. After a washing procedure with the supplied buffer, tissue sections were repaired for 40 min with ethylenediamine tetraacetic acid (UltraView Universal DAB detection kit, ready to use, Ventana Medical System, Inc., Hong Kong). The slides were incubated with the primary antibody for 60 min at 37 °C and then overnight at 4 °C. The primary antibodies, CAR (1:100, Rabbit origin, Abcam, Cambridge, UK) and DSG2 (1:50, Rabbit origin, Abcam, Cambridge, UK), were used. The IHC stain results are shown in Figure 1A,B. After three rinses in a buffer, the slides were incubated with the secondary antibodies (UltraView Universal DAB detection kit, ready to use, catalogue no. 760–500, mixture of mouse and rabbit origins, Ventana Medical System, Inc.). Tissue staining was visualized with a DAB substrate chromogen solution. Slides were counterstained with hematoxylin (37 °C, 4 min), dehydrated, and mounted.

In order to evaluate the immunostaining results objectively, we performed whole-slide scanning with computer counting. The slides were scanned with a slide scanner (Panoramic DESK II DW, 3DHISTECH, Budapest, Hungary) in 200× magnification. Computer counting software of CellQuant and PatternQuant were used (version. 2.4.0.119028, 3DHISTECH, Budapest, Hungary). Briefly, PatternQuant was trained to recognize regions of interest following the CellQuant to evaluate the H-Score. The H-score was defined by immuno-intensity, which multiplied staining percentage (range from 0 to 300). Immuno-intensity was recorded as 0 for no staining, 1 for faint staining, 2 for moderate staining, and 3 for intense staining (Figure 1C,D). The staining percentage was recorded from 0% to 100%. Both immuno-intensity and staining percentage were automatically calculated by CellQuant, which would only count region of interests recognized by PatternQuant. The original scanned slides and analyzed images were manually double checked using semi-quantitative evaluation.

### 2.5. Statistics

The IHC results for the potential correlation between clinicopathological characteristics and survival outcomes were illustrated by Cox regression analyses, multiple testing (Bonferroni) correction, and the Kaplan–Meier method. Co-expressed genes surveilled by the bi-variant correlation were measured in proliferation, apoptosis, angiogenesis, and epithelial–mesenchymal transition (EMT; E-cadherin/N-cadherin ratio), which were the calculated using Pearson correlation analysis.

## 3. Results

### 3.1. Association between Clinical Variables and Protein Expression

The clinical value of *CAR* gene expression for NL tissue advancing to AAH and to invasive LUAD presented elevated levels in concordance with various genes, as shown by gene ontology (Figure 2A,B). The heatmap demonstrates 372 expressed genes with the same or different expression to *CAR* gene (NL, AAH, LUAD) (Figure 3). The gene expression level was colored red (upregulation), black (mean gene expression), and green (downregulation). The rows represented three groups of samples: NL (blue), AAH (red), and LUAD (yellow). The columns represented individual genes. The gene name is supplied in Appendix A
Appendix A. This heatmap was drawn using the R package.

*CAR* and *DSG2* gene co-expression was higher in early stage lung adenocarcinoma and was shown to have a significantly poorer survival outcome, when compared with lower *CAR* and *DSG2* gene expression (*p* = 0.0046) (Figure 4). Based on this preliminary result, we further determined the importance of *CAR* and *DSG2* gene co-expression on LUAD survival. The correlation between CAR/DSG2 and clinicopathological features was analyzed. The demographic data for NSCLC cases from the TMA is shown in Table 1. The median follow-up period was 43.3 months. The expression level of CAR/DSG2 notably increased compared with the normal lung tissue (Figure 5A). DSG2 was especially higher in lung squamous cell carcinoma (LUSC) (Figure 5B). The two markers’ expression had no difference in early or later-stage disease. DSG2 was generally expressed at significantly lower levels in both genders and in the different histological types but was prominent in LUSC (*p* = 0.005) (Table 2). Interestingly, higher levels of CAR were present slightly more in males than in females, yet this difference was insignificant (*p* = 0.051) (Table 2).

### 3.2. Correlations between CAR/DSG2 Protein Expression, Proliferation, Apoptosis, Angiogenesis, and EMT

Since the hub expression of Ki-67, caspase 3, CD31, and EMT is commonly associated with tumor cell growth, proliferation, migration, and invasion ability (Ki-67 stands for proliferation, caspase 3 stands for apoptosis, CD31 stands for angiogenesis, and EMT ration stands for migration and invasion ability), correlations among these markers were evaluated. Our results demonstrated a strong correlation between the TMA CAR expression levels, proliferation (*p* < 0.001), apoptosis (*p* < 0.001), and EMT (*p* = 0.009) (Figure 6A). DSG2 was significantly associated with proliferation only (*p* = 0.001) (Figure 6B). The correlation between CAR and DSG2 expression was also significant (Figure 6C).

### 3.3. Survival Analysis

To evaluate the potential association between prognosis and the expression of CAR, DSG2, and the co-expression of CAR/DSG2, a Kaplan–Meier survival analysis was performed. The expression of CAR and DSG2 was divided into high and low expression levels according to the cutoff value. As shown in Figure 7, high expression of either CAR or DSG2 was associated with significantly poorer overall survival (CAR, cutoff H-score = 50.5, *p* = 0.045); (DSG2, cutoff H-score = 120, *p* = 0.001) (Figure 7A,B). Of note, the expression level of CAR and DSG2 had no impact on survival in early stage disease (Stage I and IIA) (Appendix A
Appendix A), but was significantly affected in the late-stage disease (Stage IIB, III, and IV). On the other hand, the Cox regression analysis showed that DSG2 was more dominant compared with CAR (Table 3). Furthermore, when the Kaplan–Meier survival analysis was performed to explore the co-expression of CAR/DSG2 and survival, a high CAR/DSG2 expression was associated with a significantly worse overall survival time (Figure 7C, *p* = 0.011; Figure 7D, *p* = 0.015).

## 4. Discussion

In this study, we elucidated the innovative value of *CAR* and *DSG2* gene in LUAD. Initially, GEO datasets, ClusterProfiler, STRING, TCGA gene expression data, and survminer were used to examine the survival prediction value of elevated *CAR* gene expression from NL tissue that transformed into early LUAD. There were 372 expressed genes that had the same or different expression profiles as *CAR* gene, which were illustrated by a heatmap. The concordance of the upregulated genes that expressed CAR were determined, and 16 highly associated genes, as determined by gene ontology, were screened for further analysis. This revealed that *CAR* and *DSG2* genes were expressed at >90% in 262 of the TCGA samples. This was the most important feature for survival outcome prediction. Later, we validated this finding using TMA with pathologists and a comprehensive computer counting software analysis, which assessed survival. The Pearson analysis was utilized to depict the association between co-expressed markers commonly seen in tumorigenesis, such as proliferation, angiogenesis, apoptosis, and EMT. The findings of co-expressed high *CAR* and *DSG2* gene expression with poor survival from the TCGA database were consistent with the IHC staining prediction model in the 108 TMA LUAD patients.

CAR is associated with various physiological and pathophysiological functions. CAR participates in cancer biology as a critical regulator of survival and growth. Multitumor array analysis has shown altered CAR expression in different cancer types [12]. CAR was shown to promote carcinogenesis when overexpressed in certain malignancies [7,8,9,10]. Downregulated CAR expression in NSCLC cell lines led to decreased colony formation in vitro, inhibited tumorigenicity in vivo, and fewer CSC characteristics [18,30]. In addition, silencing CAR in lung cancer cells reduced tumor engraftment efficiency as CAR exerted a potential role in EMT [31]. Since a correlation between CAR expression and CSC phenotype was established, CAR has become a potential target for CSC therapy and a predictor of treatment response in NSCLC [18].

Several previous studies have also demonstrated that DSG2 is highly expressed in several tumors, such as gastric cancer, skin squamous cell carcinomas, and melanoma [22,24,32]. DSG2 was also highly detected in primary lung cancers [25], especially in LSCC [26]. The results of the current study revealed similar findings. Zhang et al. reported that DSG2 expression was 1.5-fold higher in LUAD tissue compared with that in NL tissue, as shown by membrane proteomic analysis [33]. Similar findings were also detected by Li et al. in mitochondrial protein in LUAD patients [34]. In a recent report by Sun et al., the mRNA and protein levels of DSG2 were increased in LUAD. High expression of DSG2 was correlated with TNM stage, tumor size, and lymph node metastasis. High DSG2 levels were also associated with poor overall survival [35].

DSG2 is a cell adhesion molecule similar to CAR and is a receptor for adenovirus-based vectors. It had been widely reported that the adenovirus can be used as an oncolytic virus vector for gene therapy cancer treatment [28]. However, successful amplification of adenoviral vectors in tumor cells remains challenging [36]. A previous study proposed that viral transduction efficiency might be influenced by innate or pre-existing immune responses. Antibody–capsid chimeras or modification of specific peptides can improve host molecule specificity [37].

Taken together, increased levels of CAR and DSG2 could help target oncolytic adenovirus vectors for use as a therapeutic tool in lung cancer [5,21]; however, they also appear to indicate inferior survival outcomes in NSCLC [6,7,9,12,18,25,26,35]. Based on this theoretical assumption, we then used *CAR* and *DSG2* gene in combination as a model independent prognostic predictor for LUAD. Firstly, surveillance from GEO enrichment analysis and gene ontology listed the dominant genes categorized from molecular functions and cellular components, which were selected for mapping with the TCGA database. This generated 15 genes predominantly related to *CAR* gene (*DSG2, OCLN, CLDN4, CLDN3, KRT8, CDH1, EPCAM, PTK7, AQP3, S100A11, PERP, PTPRK, PKP2, DSC2, PAK1*). We found that *DSG2* gene was the most important. When expressed with *CAR* gene at >90%, the survival prediction was statistically significant.

Later, we examined CAR and DSG2 expression from a paraffin-embedded TMA. When compared with the NL tissue, CAR and DSG2 both increased in intensity and percentage of tissue stained among the specimens. Subsequent analysis determined the H-score was defined as immuno-intensity multiplied by staining percentage (range from 0 to 300). The cutoff H scores for CAR and DSG2 were 50.5 and 120, respectively. CAR and DSG2 had a positive correlation in the bi-variant correlation analysis (*p* = 0.002). The co-existing marker also indicated a strong association with cell proliferation, as opposed to apoptosis, angiogenesis, or EMT. Our findings suggest that CAR and DSG2 participate in cell–cell junctions, which is the linkage between cell membranes that regulates tissue homeostasis in certain mechanisms, such as tissue barrier integrity, cell proliferation, and migration.

In order to evaluate the association between CAR, DSG2, co-expression of CAR/DSG,2 and survival prognosis, a Kaplan–Meier survival analysis was conducted. Increased levels of CAR or DSG2 were associated with poorer overall survival in all stages up to 250 months, although high DSG2 expression played a much more significant role, as determined by the Cox regression analysis (HR: 6.663, *p* = 0.008). When the two markers were taken into consideration and divided into high or low expression levels according to cutoff values, high CAR/DSG2 co-expression could predict worsening outcomes both in the long term (250 months) and the short term (60 months). Other than early stage disease (stage I and IIA), CAR/DSG2 co-expression could hamper survival outcomes when disease staging was relatively advanced. We proposed that a possible mechanism might be associated with enhanced cell proliferation or EMT. Thus, our findings suggest that CAR/DSG2 expression is a comprehensive predictor of overall survival in LUAD. This could be beneficial in clinical practice and could play an important role as a potential therapeutic target for factors linked with LUAD survival.

There are several inevitable limitations in this study. First, we used public databases as screening tools. Moreover, integrated genes derived from microarray datasets analysis with next-generation-sequencing data were used. Insufficient numbers of specimens may have led to investigation bias. Second, there was a limited number of tumor-adjacent normal tissues from TMA. IHC staining resulted in normal lung tissues, which should be further verified. The relatively small sample size of TMA may have been biased toward the Cox regression analysis results. Third, future experimental studies are warranted because this study lacked a corresponding laboratory survey in vitro and in vivo. Confirmation must be conducted to validate the potential mechanisms. Further studies on an oncolytic adenoviruses (OAd) -dependent host molecule interaction, virus tropism, antibody–capsid chimera engineering, modification of antigen-specific targeting peptides, OAd transduction-sensitive or refractory, would help optimize the corresponding vectors.

## 5. Conclusions

The current study showed that CAR, DSG2, or a co-expression of CAR and DSG2 in LUAD is a predictor of overall survival. High CAR expression was positively associated with cell proliferation, apoptosis, and EMT; while DSG2 was only positively correlated with proliferation. No confounding factors were influenced by age, gender, or histologic type. LUAD patients with high CAR or DSG2, and especially CAR and DSG2, had significantly reduced overall survival rates. The usage of CAR/DSG2 combined with IHC staining had a potential clinical value as a predictor of LUAD patient survival. Further in vitro and/or in vivo studies are warranted to confirm the clinical usage concept of tumor gene delivery therapy. Combination therapy using OAd for treating cancer is vital for virus uptake and gene transfer. In summary, CAR combined with DSG2 expression could be a robust biomarker for predicting prognosis and evaluating the efficiency of gene therapy. Co-expression and enhancement could promote cellular sensitivity to adenovirus infection. Therefore, additional studies are needed to elucidate the potential mechanism of CAR/DSG2 as an optimal target for viral vectors as part of gene therapy.

## Figures and Tables

**Figure 1 jcm-09-03693-f001:**
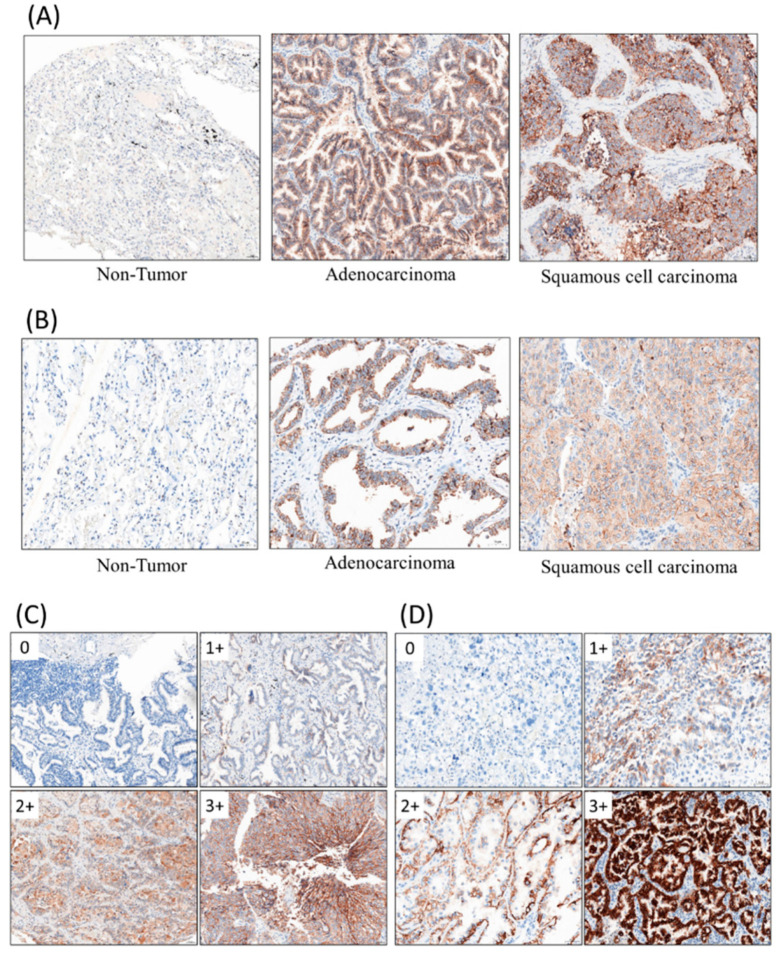
Immunohistochemical staining (IHC) stains with primary antibodies against coxsackievirus/adenovirus receptor (CAR, 1:100), desmoglein 2 (DSG2, 1:50). (**A**) CAR, (**B**) DSG2, (**C**) immuno-intensity of CAR, (**D**) immuno-intensity of DSG2.

**Figure 2 jcm-09-03693-f002:**
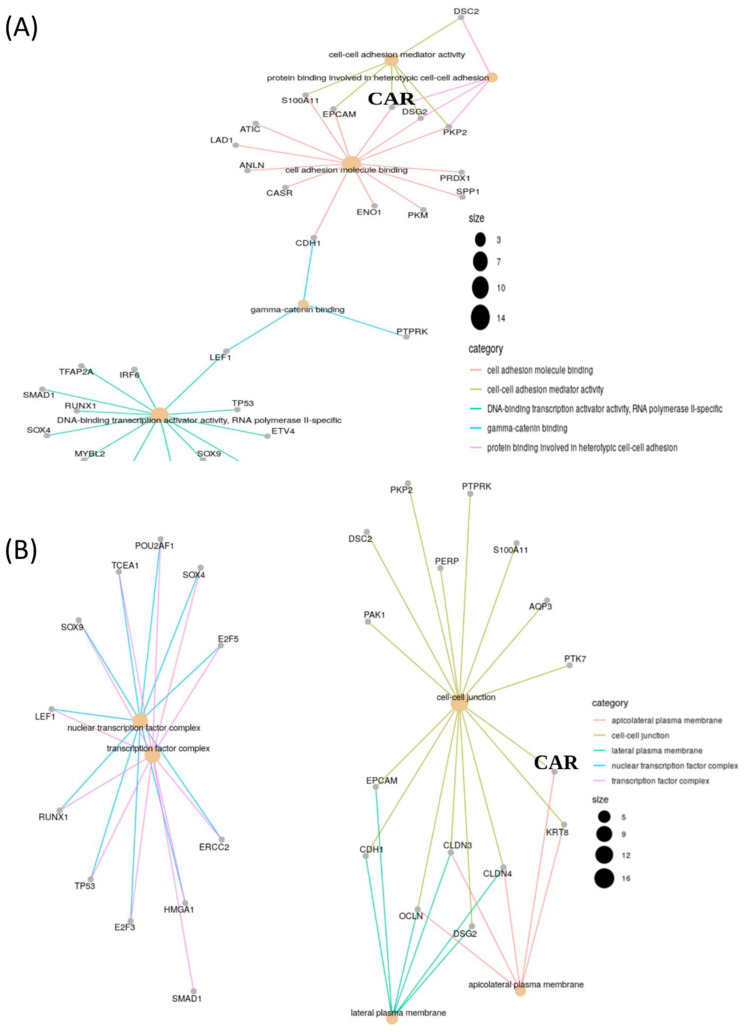
Clinical value of *CAR* gene expression in lung adenocarcinoma (LUAD) that presents an elevated level in concordance with various genes, as shown by gene ontology. (**A**) Molecular function and (**B**) cellular component.

**Figure 3 jcm-09-03693-f003:**
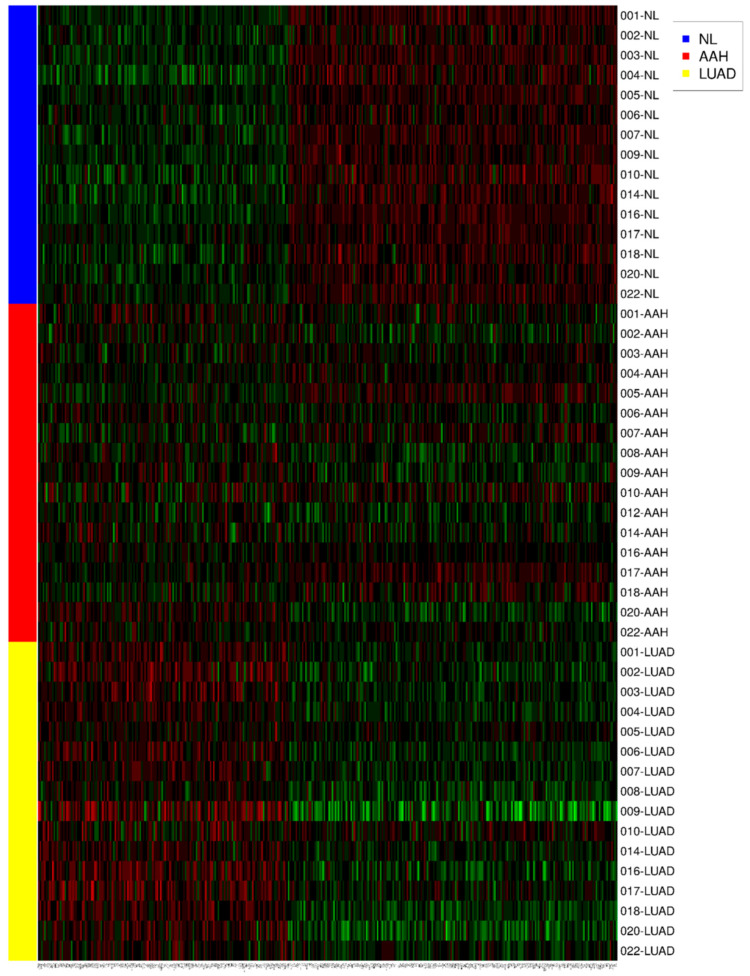
The heatmap demonstrates 372 expressed genes with the same or contrary trend as *CAR* (NL, AAH, LUAD). The gene expression level was colored red (upregulation), black (mean gene expression), and green (downregulation). The rows represent three groups of samples: normal lung parenchyma (NL, blue), atypical adenomatous hyperplasia (AAH, red), and lung adenocarcinoma (LUAD, yellow). The columns represent individual genes. This heatmap was drawn using the R package.

**Figure 4 jcm-09-03693-f004:**
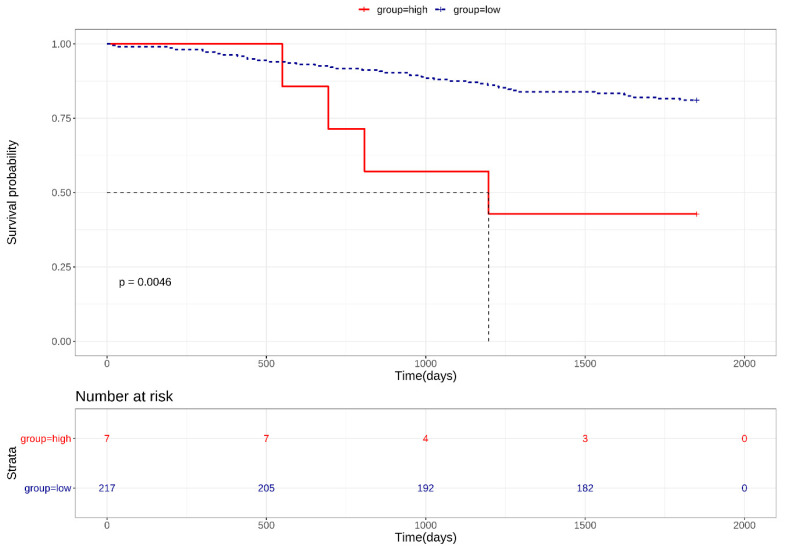
Kaplan–Meier survival curves for high *CAR/DSG2* gene co-expression in early stage LUAD.

**Figure 5 jcm-09-03693-f005:**
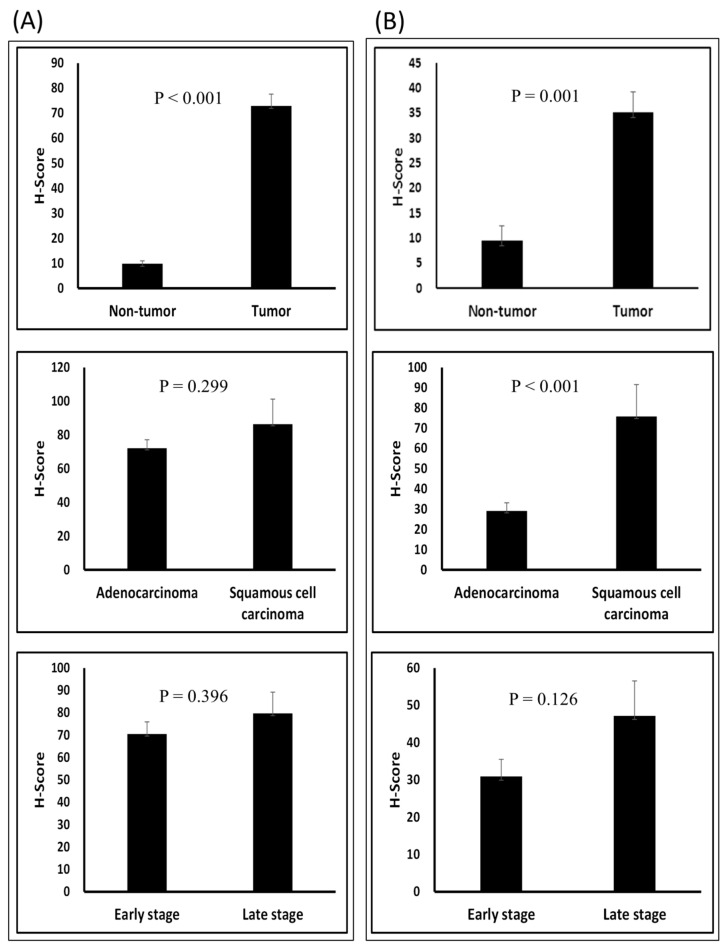
T-test of the H-score in different groups: (**A**) CAR and (**B**) DSG2 in tumor vs. normal tissue; adenocarcinoma vs. squamous cell carcinoma; and early stage vs. late stage.

**Figure 6 jcm-09-03693-f006:**
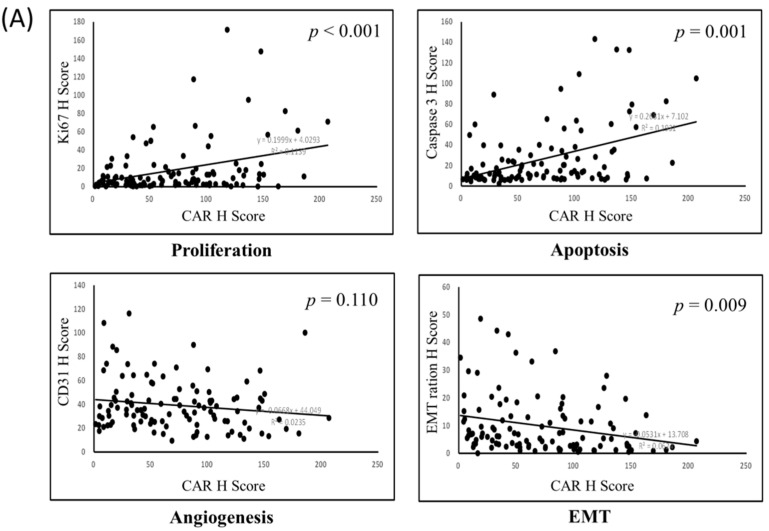
Bi-variant correlation for (**A**) proliferation, apoptosis, angiogenesis, and epithelial–mesenchymal transition (EMT) with CAR; (**B**) proliferation, apoptosis, angiogenesis, and EMT with DSG2; and (**C**) the correlation of CAR with DSG2.

**Figure 7 jcm-09-03693-f007:**
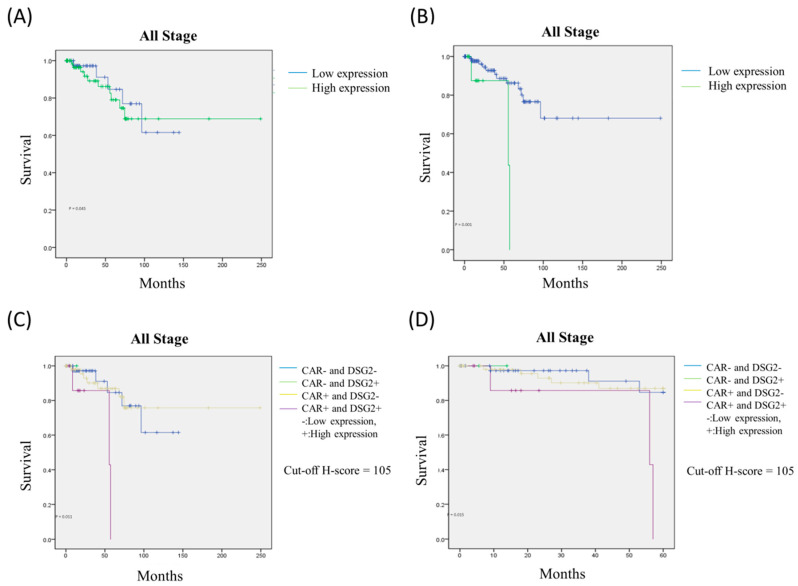
The Kaplan–Meier survival analysis for (**A**) CAR, (**B**) DSG2, and high CAR/DSG2 expression in (**C**) 250 months and (**D**) 60 months.

**Table 1 jcm-09-03693-t001:** Demographic data for non-small-cell lung cancer (NSCLC) cases.

Factors	Number or Average
N	108
Age	67.1 ± 11.2
Gender	
Male	63
Female	45
Histological type	
Adenocarcinoma	87
Squamous cell carcinoma	15
Other types *	6
Stage	
I	66
II	22
III	13
IV	7

* Mucinous adenocarcinoma, adenosquamous carcinoma, sarcomatoid carcinoma.

**Table 2 jcm-09-03693-t002:** Relationships among differentiation and clinicopathological variables in advanced lung adenocarcinoma patients.

	CAR Expression		*p* Value	DSG2 Expression		*p* Value
	Low	High		Low	High	
Gender						
Male	21	43	0.051	54	10	0.024
Female	24	22		45	1	
Histological Type						
Adenocarcinoma	36	51	0.788	81	6	0.005
Squamous cell carcinoma	4	11		10	5	
Others	3	3		6	0	
Stage						
Early	32	48	0.565	74	6	0.148
Late	11	17		23	5	

**Table 3 jcm-09-03693-t003:** Cox regression analysis of CAR vs. DSG2.

		Univariant Analysis		Multivariant Analysis			
				Model 1		Model 2 *	
Risk Factors		Hazard Ratio (95% CI)	*p* Value	Hazard Ratio (95% CI)	*p* Value	Hazard Ratio (95% CI)	*p* Value
CAR	Low	Reference	---	Reference	---	Reference	---
	High	1.297 (0.441–3.818)	0.637	1.017 (0.327–3.162)	0.976	1.065 (0.322–3.521)	0.092
DSG2	Low	Reference	---	Reference	---	Reference	---
	High	6.703 (1.746–25.738)	0.006	6.663 (1.640–27.075)	0.008	5.700 (1.161–27.981)	0.032
Age	<70 years	Reference	---			Reference	---
	>70 years	0.772 (0.261–2.283)	0.640			1.302 (0.390–4.345)	0.667
Gender	Male	Reference	---			Reference	---
	Female	0.844 (0.303–2.349)	0.075			1.252 (0.361–4.340)	0.723
Stage	Early	Reference	---			Reference	---
	Late	5.316 (1.885–14.992)	0.002			4.824 (1.666–13.968)	0.004

* The Bonferroni correction method was used to adjust the hazard ratio.

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
