# Peer review of "Co-Expression of Coxsackievirus/Adenovirus Receptors and Desmoglein 2 in Lung Adenocarcinoma: A Comprehensive Analysis of Bioinformatics and Tissue Microarrays"

_jcm, 2020, doi:10.3390/jcm9113693_

Round 1
Reviewer 1 Report
I have read with a pleasure a manuscript entitled “Co-expression of coxsackievirus adenovirus receptor and desmoglein 2 in lung adenocarcinoma: a comprehensive analysis by bioinformatics and tissue microarray". The text is well written. It raises an important issue of recently identified receptors expressed on tumor cells that influence lung cancer survival. I have significant remarks concerning this paper and I urge authors to reply to them before publication.
Major remarks:
- Please elaborate more on study samples. It must be stated more extensively what was the methodology of the study. Please do not be more detailed but rather try to present a general concept. What do the data sets consist of? How was the analysis performed? The text must be clear not only to narrow specialists in the field but also to clinicians reaching beyond daily routine (lines 89-114).
- The ethical aspects are presented in lines 123-126. It is not mentioned whether the study was approved by the Ethics Committee.
- What kind of samples were included? Did the authors use surgical specimens/biopsy specimens? Were the samples reassessed to confirm again the type of NSCLC?
- Was the follow-up assessed for every patient?
- What was the median time of follow-up?
- Please define the meaning of the early and late stages in your study.
- Figure 7. It is difficult to compare the survival of tested factors in a heterogeneous population of different NSCLC stages. Consider the comparison of prognosis in a less heterogeneous population i.e. stage I+II.
- Consider multivariable analysis to confirm the independent significance of novel molecular factors and their influence on survival.
Minor remarks:
- LUAD, LSCC are not commonly recognized abbreviations and are misleading. I suggest the use of AC, SCC.
- NL, AAH abbreviations need to be explained in an appropriate place in the text.
Author Response
Thanks very much for your kindly suggestion, we had amend article as your comments and send for English editing by MDPI. Please see the attach file!

Reviewer 2 Report
JCM967532
This is a poorly written manuscript, difficult to follow, lacking rational for method and approach chosen.
Introduction
Nice summary of CAR and DSG2
The statement˜: ‘To date, there has 77 been little research into the diagnostic and prognostic value of DSG2 in NSCLC’ is puzzling as no info about diagnostic value has been cited. Subsequently a manuscript is cited for poor prognosis in pulmonary adenocarcinoma.!!??
The aim of the study: ‘Therefore, the current study investigated the use of CAR in combination with DSG2 as predictors of LUAD outcomes.’ Why this combination is not clear.
Methodology
4 on line available data sets were examined for CAR and DSG2, which seems to be highly expressed in 90% of 200+ samples.
A commercial TMA was used [n=108 ‘lung cancer’ n=2 normal]. Was this a sequential series of cases or cases with larger tumors and remaining fragmenst for commercial purpose?
IHC is performed for CAR and DSG2.[limited info about IHC method provided], slides scanned and H-score extracted. No info about reproducibility of this approach.
The expression of 210 CAR and DSG2 was divided into high and low expression levels according to which cut-off value??
Last sentence: ‘Correlation were measured in proliferation, apoptosis, angiogenesis and epithelial-mesenchymal transition (EMT; E-cadherin/N-cadherin ratio), which were calculated by Pearson correlation analysis.’ No info how ‘proliferation, apoptosis, angiogenesis and epithelial-mesenchymal transition (EMT; E-cadherin/N-cadherin ratio’ data is obtained, rational?
Results
Figure 2. Clinical value of CAR expression in LUAD presenting elevated level in concordance with 184 various genes as shown by Gene Ontology. (A) Molecular function, (B) cellular component. There is not info about CXADR, emphasized on the image. Content: what is the clinical value of these ontology derived distributions?
Figure 3. Selection criteria of the subjects.’ There are no selection criteria shown.
Explanation for nl , aah and luad is lacking
Figure 4 Survival curves: group high and group low are not defined.
Time has no unit {days or months etc?)
Numbers are in one group very very small: 7 compared to the other 217 in the other group
Figure 5: it is unclear which figure is associated with CAR and which with DSG2: The A, B and C are not matching. ‘C’ is shown in legend, but not shown in the figure.
Table 1
Histological types: other type has *, but not explanation for *
Statistics
Multiple testing (Bon-Ferroni) correction is lacking
Discussion
First sentences: not main message but more detailed than results.!!
‘In this study, we first elucidated the innovative value of CAR and DSG2 in LUAD. Initially, 223 GEO datasets, ClusterProfiler, STRING, TCGA gene expression data and Survminer were used to 224 examine the survival prediction value of elevated CAR expression from NL tissue that transformed 225 into early LUAD. There were 372 expressed genes that had the same or different expression profiles 226 as CAR, which was illustrated by a heatmap.’ Only 372 genes examined??? According to the sentence this should be true for all examined genes….
Minor
Abstract has no conclusion
CAR is a potential marker of cancer stem cells (CSCs) and caused drug-resistance in 67 NSCLC. Citation is missing
Author Response
Thanks very much for your suggestion, we had amend article as your comments and send for English editing by MDPI. Please check the attach file.

Round 2
Reviewer 1 Report
Dear Authors,
I appreciate the efforts you have made to comment on my remarks. I think that you did your best and I recommend releasing the paper to the public.
Thank you very much, it was a pleasure for me to be acquainted with the manuscript.
Author Response
Dear Reviewer:
I appreciate so much for your comprehensive review and kindly comments. This help us improving our article significantly. Thank you so much and it was also our honor to submit this manuscript to Journal of Clinical Medicine.
Sincerely,
Ching-Fu Weng, MD